# The Determinants of Income of Rural Women in Bangladesh

**Md.Shajahan Kabir [1], Mirjana Radović Marković [2,3,\*] and Dejan Radulović [4]**

[1] Department of Rural Sociology, Bangladesh Agricultural University, Mymensingh 2202, Bangladesh; shajahan.rs@bau.edu.bd
[2] High School of Economics and Management, South Ural State University, Chelyabinsk 454080, Russia
[3] Department of Basic Research, Institute of Economic Sciences, Belgrade 11000, Serbia
[4] Faculty of Law, Business Academy University, Novi Sad 21000, Serbia; radulovic@notard.rs
[\*] Correspondence: mirjana.radovic@ien.bg.ac.rs

**Abstract:** This study investigated the factor which influences rural women's income after participation in small-scale agricultural farming, their contribution to the household, as well as their empowerment status. This research was conducted in Jinaigati upazila of Sherpur district in Bangladesh. A total of 80 respondents (women) from this upazilla of Sherpur were selected purposively using simple random sampling. The quantitative data were collected by in depth interviewing of the 80 respondents through personal interview. The quantitative analytical tools used to attain specific objectives included various descriptive statistics, functional analysis, multiple regression co-efficient, used to identify the factors of influencing women's income through small-scale agricultural farming. Problem Confrontation Index (PCI) used through different problems identified scores. In accordance with the results of the educational level of woman, other sources of income, experience and training, access to credit, decision-making ability have an positive impact on rural women's income. These variables were statistically significant. From the Problem Confrontation Index, it was found that lack of capital was the first ranked problem, need-based training the second ranked problem, high interest rate the third ranked problem, insufficient farm size the fourth ranked problem, and lack of quality of seed the fifth ranked problem. Their income from this brought remarkable positive change in their life and they had better control over their decisions and income. Finally, their active economical participation in small-scale farming assists them to overcome prejudice, socio-economic barriers, and highest empowerment attainment in the context of Bangladesh—and, if the government takes proper initiative in terms of gender policy, then rural women's income and livelihood status will be increased remarkably.

**Keywords:** rural women; agricultural farming; sustainability

## 1. Introduction

Women's well-being is determined by self-esteem and harmony in relationships. Radović-Marković, Nelson-Watchman, and Omolaja [1] defined entrepreneurship as the formation of entrepreneurial affiliations, which plan to guarantee the improvement of crucial economic, social, and different changes in rural areas. It would be achieved through people making innovations and governmental strategy considering putting resources into rural entrepreneurship. This means that meaningful agricultural development could be achieved by identifying those barriers militating against the participation of women in agricultural development. However, agricultural research activity and extension services neglect women farmers who are facing a lot of problems that, to a large extent, limit their potential in agricultural and entrepreneurial development [2].

Women commonly face higher risks and difficulty from the impact of climate change in situations of poverty. "Scientific evidence on climate change can by no means be disputed, however, it is difficult to predict the way in which this change will affect certain regions and countries" [3] (p. 56). So, this effects their limited adaptive capacities and their dependency on climate-sensitive resources [4]. Therefore, several researchers recommend "that smallholder farmers in developing countries may combat climate change by returning to more natural productive systems, which provide improved ecological and social features" [5] (p. 6). Also, the latest research draws attention to women needing to gain better knowledge regarding appropriate adaptations to climate changes, recent technologies, and availability of credit and loan facilities [2,6–9].

Although there is no clear relationship between gender equality and sustainability, both make an impact on economic development [10,11]. In addition, improving environmental sustainability will likely ensure a positive impact on the rural households and can be the major force for generating agricultural productivity [12]. Dankelman and Davidson [13], in their research, concluded that women have the potential to be the bearer of change in conserving natural resources, and in making a key contribution to environmental recovery. At the household level, women and men are responsible for different specialized activities. "Many of these activities are not considered as *economically active employment*, but they are all critical to the well-being of rural households" [13] (p. 2). In this context, gender roles vary enormously across societies and cultures. Power imbalances and gender differences will likely create winners and losers among them [14].

Women and men differ from each other in the roles they play in agriculture, e.g., roles in the cropping cycle or ownership of crops or livestock. The position of women in agriculture is much more precarious compared to men regarding their poorer working conditions. For example, women are paid less than men for equal work. Women's businesses in agriculture are generally smaller in terms of size, turnover, and number of employees. Agriculture policy measures should expand income earning opportunities and secure recognition of women's work that is otherwise under-valued, as well as support women's access to financial and other resources such as land, labor, and water.

"Gender equality for women farmers starts with recognizing their contribution to sustainable food production and supporting their priorities" [15] (p. 1). Women are empowered in the agricultural sector when they can manage resources and make decisions about food production and income. In other words, access to resources is of key importance for gender equality [16].

It should be emphasized that in developing countries, women typically face a range of risks that include environmental, economic, and social obstacles [17]. For example, in Pakistan, Afghanistan, Bangladesh, and India, considering formal, standard, or religious laws, women are contrarily impacted with respect to the rights to possess property [18]. So, women in agriculture represent silent workers who have a large but under-valued contribution to the rural economy [19].

To ensure balanced socioeconomic development of the country, emancipation of women with changed social status is a precondition which can be achieved only through increased paid employment of women and by utilizing their talents for national development. It is not unattainable to turn these largely unemployed, unskilled, and poor rural women into a weapon for improving economic growth. Poverty, inequality, illiteracy, unemployment, disempowerment, and food insecurity in the case of rural women in Bangladesh have appeared to be severe glitches for a long time. It is possible to overcome these glitches by employing rural women in their most suitable employment sector—agriculture. In this context, small scale agriculture such as livestock and poultry, vegetable farming, and petty enterprises are involved in the planned strategy for providing balanced development of the economy of poor women in Bangladesh. The nature of agricultural employment in the case of women in Sherpur district has changed a lot. Women in this district have been engaged with agriculture—mostly as family labor—for a long time, but due to happenings like rapid urbanization and industrialization, swift out-migration of men, government policies and initiatives, etc., some labor scarcity has been experienced here. So, this study aims to find out and analyze the impact (empowerment and role) of small-scale agricultural farming of rural women's participation in Sherpur district.

In this paper, the specific objectives are:

1.    To identify the factors influencing rural women's income in small-scale agricultural farming.
2.    To find the major barriers of rural women and their suggested recommendation in their areas.

This analysis was achieved via the review of literature on women's participation in small-scale agricultural farming or empowerment of rural women through income-generating activities. This study tried to investigate rural women's involvement in small-scale agricultural farming and its effect on their household.

## 2. Theoretical Overview

The government of Bangladesh has made a long-term great effort to reduce poverty and malnutrition [20]. Furthermore, in Bangladesh has been insisted on green agriculture during the past two decades [21]. Consequently, small farming can play a key role in providing enough food supply and help to improve nutrition security in rural areas [22]. This implies that small farming can contribute to family income and diversified food products. Relying on data, over 80% of the country's 14.7 million agriculture farm households are smallholders who wield less than a quarter of a hectare of cultivated area and typically practice intensive subsistence agriculture [23]. In addition, women's participation in agriculture has grown exponentially in this last two decades. However, women face serious difficulties in their work in the agricultural sector. Namely, they must work under poor conditions, defying social barriers and discrimination. Poverty, illness, and negligence are their common allies. Namely, rural women are the worst sufferers both at home and outside in society. So far, much has been talked about regarding gender equality. Jaim and Hossain [24] used the term "Feminization of agriculture" to describe the changing scenario. FAO [25] reported that women comprise about 43% of the global agricultural labor force and of that in developing countries, but this figure masks considerable variation across regions and within countries according to age and social class. Khatun and Kabir [26] studied the topic of ensuring women's empowerment in Bangladesh through entrepreneurship. They stated that the social, political, and economic conditions of women are very vulnerable. Women who are not permitted to go outside can also run their business in their home by making cakes, tailoring, gardening, poultry, fishing in their family ponds, and so on. Now, their contribution to economic growth and employment is noticeable and the above-mentioned reviews represent small-scale agricultural farming, women's role and status in agriculture, their contribution to the family and overall economy, and the status of women's empowerment. Income generation through women's participation has become a global issue in current time, as well as a great concern for the future [27]. Again, the increasing high population needs higher participation of human resource in the work force, which results in income generation and empowerment of women. In this connection, the present study attempted to conduct an in-depth study that can capture women's contribution to their family income through agricultural farming, as well as their empowerment situation, in a selected district of Bangladesh. Attention has also been given to draw out the problems and challenges faced by the women in that area.

Training is an essential component for human growth and employment [28,29]. Despite Bangladesh's effort, a moderate change has been made in the areas of female education. However, this small change and improvement appears to be very insignificant in mitigating the suffering of the female work force. Participation of female labor compared to male labor has increased in the agricultural sector over the period 2000-2018. During same period, the female labor force has increased by 136.025%, which is much higher than the male labor force's increase of 35.633%. In the case of the agricultural sector, where female employment has increased by 192.84%, there has been a sharp decline in the participation rate for males by 16.26%. While there were only 3.8 million women engaged in agricultural employment during 1999–2000, the number was 18.646 million in 2016–18, which can be observed from Table 1.

**Table 1.** Changes in employment by gender.

|  | 1999–2000 | 2016–2017 | Change | 1999–2000 | 2016–2018 | Change |
| --- | --- | --- | --- | --- | --- | --- |
|  | Agriculture (in '000') | | (%) | Total labor force (in '000') | | (%) |
| Male | 16,200 | 13,565 | −16.26 | 31,100 | 42,182 | 35.633 |
| Female | 3800 | 11,128 | 192.84 | 7900 | 18,646 | 136.027 |
| Total | 20,000 | 24,693 | 23.46 | 39,000 | 60,828 | 55.97 |

Source: Bangladesh Statistics 2016.

Despite their domestic work, rural women in Bangladesh now play an active role in providing household food security. They do not only ensure protein supply for the family by rearing livestock or poultry, but also contribute to household diet by growing various vegetables and fruits in the homestead garden. Farm activities in the homesteads, ranging from selection of seed to harvesting and storing of crops, are predominantly managed by women. Half of the population of Bangladesh is female, and their economic participation has increased significantly. To attain the goals initiated by the Bangladesh government for women's development, the country has approved the highest allocation in the budget for the 2018–2019 fiscal years. Bangladesh considers women's participation a vital issue in the path of women's empowerment and one of the main drivers of transforming the country's status from low-income to middle-income. Women's advancement through access to education [30], health, the labor market, employment, and social protection were prioritized in the financial year 2019 (FY19) budget, which is around 30% of total budget size. Samira Zuberi Himika, managing director of GIGA TECH and founder of Team Engine, said to Dhaka Tribune: "Mobility and combination of possibilities are extremely influential for women and girls to strengthen their position among the working population." Participation of girls in primary schools is increasing, as their overall enrollment rose from 57% in 2008 to 95.4% in 2017 [31].

In Bangladesh, being a traditional Muslim society, women harshly participate in agricultural activities outside the home [32]. Women's agricultural activities were traditionally confined to homestead production and post-harvest operations. However, in recent years, they are mostly involved in livestock and poultry rearing activities, besides crop production.

A few studies were conducted on women's activities in Bangladesh. Mostly, these studies showed that females have very limited control over land and the household resources, and that their voice is very rarely valued. Therefore, participation in agricultural farming by women can be considered as an important engine of engaging women in economic activities [33].

*The Conceptual Framework of the Study*

Several socioeconomic aspects of the samples were examined, such as average family size, level of education, farm size, training and experiences, access to credit, and so forth (independent variables). Because the successful selection of variables results in success of the research, the authors of the study employed adequate care in selecting the variables.

In light of the foregoing discussion, a conceptual framework has been developed for this study, which is diagrammatically shown in the Figure 1.

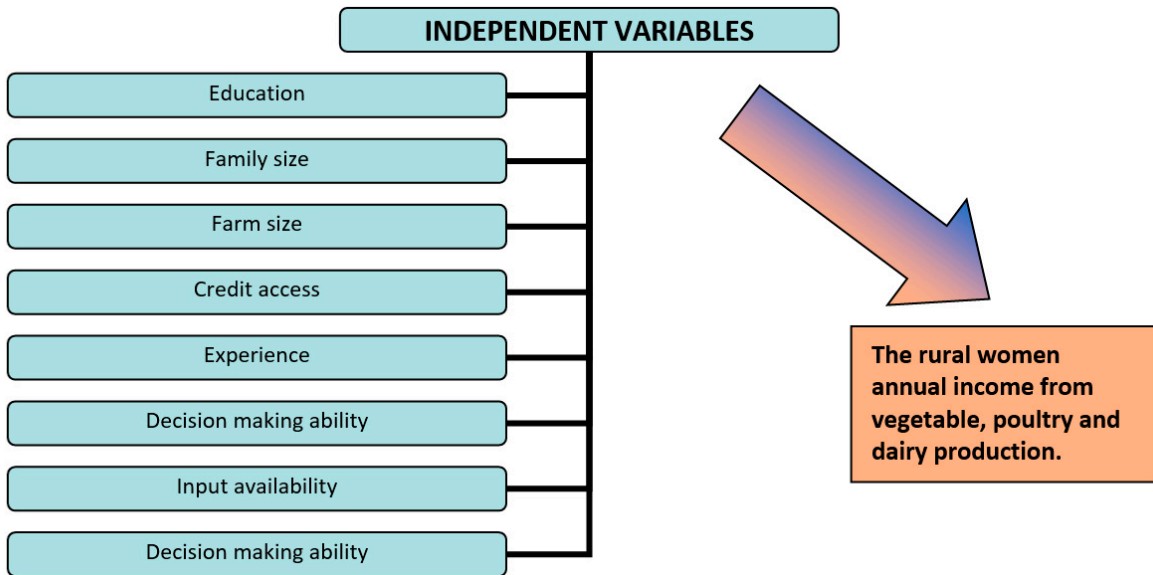

**Figure 1.** A simple conceptual framework for the study. Source: Authors.

Rural women's annual income from vegetable, poultry, and dairy production was the focus of this study and considered the dependent variable.

## 3. Method

The present research work was carried out in Jinagati upazila of Shepur district, which belongs to Mymensingh division in Bangladesh. The location of the selected study area is presented in Figure 2 below.

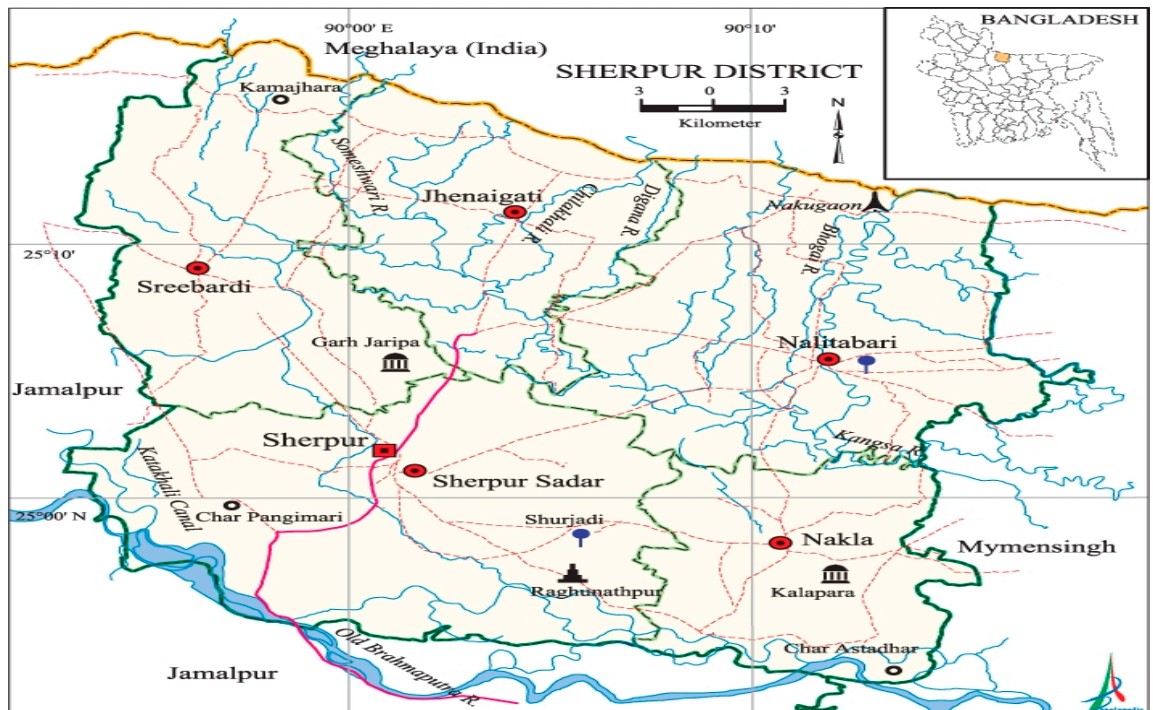

**Figure 2.** Location of the study area.

For necessary comparisons, a total of 80 respondents involved in small-scale agricultural farming under the category of homestead gardening, poultry, and dairy farming were interviewed.

Initially, 25 respondents engaged in homestead gardening, 25 respondents engaged in poultry, and finally 30 respondents engaged in dairy were selected purposively by using simple random sampling techniques.

## 4. Instrument for Data Collection

The data were collected during the period of June to July 2018. Both qualitative data on the general perspectives of the people studied and quantitative data that allowed exploring specific issues in which the researcher was interested were collected. Primary data were collected by the researcher through face-to-face interviews by making personal visits by the researcher, both at the home and at the workplace of the respondents. Before beginning the interview, each respondent was given a brief idea about the nature and purpose of the study, and they were convinced that it was solely for academic research purposes. Both closed and open questions were included in the interviews. Answers were recorded carefully.

*Data Collecting Procedure and Statistical Technique*

Data entry was done on a computer by the researchers themselves. After completion of the field survey, data collected were coded, tabulated, and analyzed in accordance with the objectives of the study.

Narrative and descriptive techniques were followed to analyze the collected data by using the concerned software, such as SPSS and Microsoft Excel.

Descriptive analyses, such as range, number, percentage, mean, standard deviation, and rank order, were used whenever possible. Pearson's product moment co-efficient of correlation (r) was used in order to explore the relationship between the concerned variables. Throughout the study, at least a 5% (0.05) level of probability was used as the basis for rejecting a null hypothesis.

In this research, the multiple regression co-efficient was used to determine the different factors and to identify the factors influencing women's income through small-scale agricultural farming.

The model was as follows:

$$Y_i = \beta_0 + \beta_1 \cdot X_1 + \beta_2 \cdot X_2 + \beta_3 \cdot X_3 + \beta_4 \cdot X_4 + \beta_5 \cdot X_5 + \beta_6 \cdot X_6 + \beta_7 \cdot X_7 + \beta_8 \cdot X_8 + \varepsilon_i \tag{1}$$

where,
$Y_i$ = Income of rural women
Independent variables:
$X_1$ = Level of educational attainment (years of schooling)
$X_2$ = Family size (number)
$X_3$ = Farm size (acre)
$X_4$ = Others sources of income (amount)
$X_5$ = Input availability (yes or no)
$X_6$ = Access to credit (amount)
$X_7$ = Decision making ability (yes or no)
$X_8$ = Training and experiences (number of days)
$\beta_0$ = Intercept
$\beta_1$ to $\beta_8$ = Regression co-efficients of the independent variables
$\varepsilon$ = Disturbance term or error term

Correlation co-efficient (r) was performed to determine the relationship between various factors (education, family size, farm size, other sources of income, input availability, training received, training and experiences, access to credit, decision-making ability). Here, the dependent variable was rural women's annual income from vegetable, poultry, and dairy production (Figure 1).

The Problems Confrontation Index (PCI) is a measure of determining problems and constraints where problems are shown in tabulated form according to their severity. By using a structured

questionnaire, the women were asked to give their opinion on selected problems during data collection. Following the methodology used by Ismat et al. (2009), "Livelihood improvement of small farmers through family poultry in Bangladesh" PCI has been calculated. A four-point rating scale was used for computing the problem score of a respondent. The respondents were given four alternative responses ("high, "medium, "low", and "not at all") for each of the eight selected problems. Scores were assigned to those alternative responses as: "High" = 3, "Medium" = 2, "Low" = 1, and "Not at all" = 0, respectively. The Problem Confrontation Index (PCI) was computed by using this formula: Problem Confrontation Index (PCI) = Ph × 3 + Pm × 2 + Pl × 1 + Pn × 0, where Ph = total number of rural women that expressed "high" problems; Pm = total number of rural women that expressed "medium" problems; Pl = total number of rural women that expressed "low" problems; and Pn = total number of rural women that expressed "not at all" problems.

## 5. Key Findings and Discussion

Participation of rural women in income-generating activities is not an unfamiliar issue in Bangladesh. Most of the women in rural areas are directly involved in agricultural activities, which may be recognized or not. There were a number of factors that determined the involvement of rural women in influencing income through small-scale agricultural farming (poultry, dairy, vegetable production), which are discussed here.

**Specification of the model:** Specification is the process of developing a regression model. This process consists of selecting an appropriate functional form for the model and choosing which variables to include.

**Effects of selected factors on income earning:** The regression result (estimated values of the co-efficient and related statistics) is presented in the following table, Table 2.

**Table 2.** Factors affecting income in the studied households.

| Parameters | Co-Efficients | SE [1] | *t*-Stat [2] | *p*-Value [3] |
|---|---|---|---|---|
| Intercept | −0.55 | 0.37 | −1.46 | 0.15 |
| Education of respondent ($X_1$) | 0.05 | 0.02 | 2.50 | 0.01 *** |
| Family size ($X_2$) | 0.034 | 0.05 | 0.68 | 0.16 |
| Farm size ($X_3$) | −0.03 | 0.02 | 1.45 | 0.15 |
| Other sources of income ($X_4$) | 0.09 | 0.04 | 2.25 | 0.01 *** |
| Input availability ($X_5$) | 0.00 | 0.00 | 0.40 | 0.16 |
| Access to credit ($X_6$) | 0.12 | 0.06 | 2.03 | 0.05 ** |
| Decision-making ability ($X_7$) | 0.02 | 0.11 | 0.1818 | 0.24 |
| Training and experiences($X_8$) | 0.39 | 0.11 | 3.545 | 0.000 *** |
| Adjusted R Square | | 0.53 | | |

Note: *** indicates 1%, ** indicates 5%, and * indicates 10% level of significance. [1] The standard error (SE) of a statistic (usually an estimate of a parameter) is the standard deviation of its sampling distribution. [2] The t-statistic is the ratio of the departure of the estimated value of a parameter from its hypothesized value to its standard error. 3 The *p*-value is the probability of finding the observed results when the null hypothesis (H0) of a study question is true. Source: Authors' estimation, 2018.

## 6. Problem Confrontation Index (PCI)

The Problem Confrontation Index (PCI) is a measure of determining problems and constraints where problems are shown in tabulated form according to their severity. By using a structured questionnaire, the women were asked to give their opinion on 12 selected problems during data collection. Following the methodology used by Ismat et al. [34], a four-point rating scale was used for computing the problem score of a respondent. The respondents were given four alternative responses as "high", "medium", "low", and "not at all" against each of the eight selected problems. Scores were

assigned to those alternative responses as: "High" = 3, "Medium" = 2, "Low" = 1, and "Not at all" = 0, respectively. The Problem Confrontation Index (PCI) was computed by using this formula: Problem Confrontation Index (PCI) = Ph × 3 + Pm × 2 + Pl × 1 + Pn × 0, where Ph = total number of rural women that expressed "high" problems; Pm = total number rural women of the expressed "medium" problem; Pl = total number of rural women that expressed "low" problems; Pn = total number of rural women that expressed "not at all" problems.

The regression co-efficient of the education of respondents was 0.05—statistically significant at the 5% level. This implies that holding all other variables constant, a 1% increase in educational level of the respondents would lead to an increase of income by 5%. This indicates that rural women's education strongly affects their income. Several studies in Bangladesh regarding education and income support our findings [25]. A survey was conducted by Mahmudul et al. [26] among farmers in Bangladesh, and they found that literate famers achieve higher income than illiterate farmers.

The results show that family size did not at all effect rural women's income in the study area. Rural women must do their farming alone or with little cooperation from their family members or engaged hired labor. As a result, rural women's income was not influenced by their family member.

The regression co-efficient of farm size was −0.03. This indicates that poor farmers' farm size is very small, so its impact on women's income of the household is negatively correlated. Another study finding shows, in line with expectations, that farm size showed a positive and significant effect on rural women's income. In rural Bangladesh, families who have a large farm are richer and they have more opportunities to earn money than families with comparatively small farms [27]. So, increase in farm size ultimately increases production, which ensures high income, as well as a better standard of living.

The regression co-efficient of another source of income was 0.09. The co-efficient was statistically significant at the 5% level. This indicates that holding all other variables constant, a 1% increase in other sources of income would lead to an increase in women's income by 9%. Other sources of income enhance supply capital formation in their small farming activities.

The regression co-efficient of access to credit was 0.12. The co-efficient was statistically significant at the 5% level. This indicates that holding all other variables constant, a 1% increase in access to credit would lead to an increase of women's income by 12%. In our national development context, micro credit plays a very vital role and influences rural women to be good entrepreneurs.

The regression co-efficient of decision-making ability was 0.02, which was not significant. The regression co-efficient of training and experiences was statistically significant at the 1% level. This indicates that holding all other variables constant, a 1% increase in training would lead to an increase of income by 0.39%. The result shows that the farming experience of women has a positive co-efficient and is highly significant at the 1% level. So, this factor reveals that women who were more experienced had higher income. With an increase of experience, there will be an increase in income. This result is in line with previous studies in Bangladesh by Rahman, where they found that experience had a significant positive effect on annual income. This means that rural women's income is greatly influenced by their training facilities. These training is normally arranged by government (GOs) or non-government organizations (NGOs), which gives them extra knowledge about farming such as inputs, marketing condition, vaccination, etc. Training facilities have a strong impact on rural women's income in the study area.

## 7. Computation of the Problem Confrontation Index

To measure the extent of the severity of the problems confronted by rural women in small-scale agricultural production, the Problems Confrontation Index (PCI) was computed. The computed PCI of the eight problems ranged from 156 to 114 against a possible range of 0 to 225. Thus, the PCI of individual problems could range from 0 to 225, where 0 indicates "no" problem confrontation and 225 indicates "high" problem confrontation.

The results have been arranged in rank order according to their problem severity, which is shown in Table 3.

**Table 3.** Computation of the Problem Confrontation Index (PCI).

| Sl. No | Problems | Extent of Problem Confrontation | | | | PCI | Rank Order |
|---|---|---|---|---|---|---|---|
| | | High (3) | Medium (2) | Low (1) | Not at All (0) | | |
| 1 | Insufficient farm size | 24 | 26 | 30 | 0 | 143 | 5 |
| 2 | Lack of quality of seed | 20 | 21 | 39 | 0 | 136 | 6 |
| 3 | High price of inputs | 27 | 23 | 40 | 0 | 151 | 3 |
| 4 | Lack of need-based training facilities | 30 | 24 | 26 | 0 | 155 | 2 |
| 5 | Transportation problem | 20 | 18 | 42 | 0 | 115 | 7 |
| 6 | Non-cooperation from husband | 15 | 25 | 40 | 0 | 114 | 8 |
| 7 | High rate of interest | 22 | 35 | 23 | 0 | 147 | 4 |
| 8 | Lack of capital | 25 | 40 | 15 | 0 | 156 | 1 |

Source: Sample survey, 2018.

## 8. Interpretation of the Problem Confrontation Index (PCI)

### 8.1. Lack of Capital

The majority of the respondents pointed out that lack of credit is a major problem in the study area. Out of 80 respondents, 25 women faced this problem to a high extent, 40 women faced this problem to a medium extent, 15 women confronted this problem to a low extent, and there were none who said that lack of credit was not a problem. In this case, the computed value of PCI was 158[(25 + (40 + 15))] against a possible range of 0 to 225.

### 8.2. Lack of Training Facilities

Training facilities enrich and upscale existing knowledge and experience. The majority of the respondents pointed out that lack of training facilities is a major problem in the study area. Out of 80 respondents, 30 women faced this problem to a high extent, 24 women faced this problem to a medium extent, 26 women confronted this problem to low extent, and there was no one who said that lack of training facilities was not a problem. In this case, the computed value of PCI was 155[(30 + (24 + 26))] against a possible range of 0 to 225.

### 8.3. High Price of Inputs

A good number of women mentioned that the higher price of inputs is a problem in agricultural production. In this case, the computed value of PCI was 151[(27 + (23+ 40))] against a possible range of 0 to 225. In the research area, most of the women possessed a very small amount of land as their own property. Being involved with small-scale agricultural farming, their return from their farming is also small. That is why most of the respondents said the high price of inputs has created a high cost of production which they felt needed to be solved by the government or other organizations.

### 8.4. High Interest Rate

It is very difficult for rural women to get credit from any organization without any collateral. Formal credit from different institutions is very difficult to get and requires complicated procedures. Therefore, sometimes they borrowed money from landlords or informal money lenders, or relatives and neighbors, against higher interest rates. The PCI value was 147[(22 + (35 + 23))], which scored the fourth largest value range from 0 to 225 of the problems in the problem index.

### 8.5. Insufficient Farm Area

Respondents pointed out that the lack of land as the eight largest rank, as they said they needed more land to increase their production. Lack of profit was ranked the sixth problem of the study area. The PCI value was 141[(24 + (26 + 30))], which scored the eight largest value range from 0 to 225 of the problems in the problem index.

### 8.6. Lack of Quality Seed

Lack of quality seed and lower production of local breed was determined as the ninth ranked problem of the problem index. The PCI value was 136[(20 + (21 + 39))] for this problem. They said because of the poor seed quality, their production has gradually decreased. They also said that because of insufficient capital, they were not able to rear high yielding breed.

### 8.7. Transportation Problems

Transportation was not a major problem in the selected areas, as the PCI value of transportation problems was 115. The selected villages were situated near to the local market and not very far from Sadar upazila. Transportation facilities were comparatively better in those areas.

### 8.8. Husband Non-Cooperation

Discouragement from their husband was described as the lowest possible problem of the specified twelve problems faced by rural women of agricultural production, with a PCI of 114 [(15 + (25 + 40))]. Most of the respondents said that their husband had never discouraged them for doing agricultural farming. This is very good to hear—that women get more or less equal opportunities like men.

## 9. Conclusions

Agricultural production is a source of employment opportunity for mostly unemployed and unpaid rural women, and can also increase the income of women, which may result in the better use of household resources and improved livelihood. When women can add income to their family, they can participate in decision-making of their family. That is a way they may be empowered.

Migration of the male members from rural to urban, the role of rural women is changing from unpaid family labor to farm managers, a phenomenon termed "Feminization of agriculture".

Rural women are now actively involved in these activities by taking a small-scale of agricultural loan from GOs and NGOs, and these rural women now act as agricultural entrepreneurs. When a rural woman works outside, she can contribute to the family income, which has a great impact in improving their livelihood status. In fact, this change can help them to also take part in family decisions, which indicates the empowerment of rural women. Although, the contribution of rural women cannot be marked as much as should be needed.

The research results from different dimensions depict that the overall position of the selected women ranged between low to medium. Conclusions drawn based on findings of this study are as presented below:

1. Various factors such as unavailability of enough credit, insufficient capital, inadequate training facilities, shortage of helping hand problem, and high price of inputs are faced by women in this area. These problems make women's work harder in livestock production.
2. The study detected that lack of financial resources had a consequential, negative relationship with the problems confronted by female farmers.
3. Rural women's income of selected households is strongly affected by their education, farm size, farming experience, and the training they have received [35].
4. This study accepted the fact that after economical participation of women in small-scale farming, most of the respondents enjoyed a relatively better position in the household and increased social prestige, which ultimately provides a peaceful life.

5.    After all, economically active contribution of the respondents in the research area gives them the opportunity to improve their empowerment status in their household.

6.    Necessary attentions are required from the respective authorities to solve these problems.

It is expected that our research will help the administrative and policy-makers of both GOs and NGOs, who are concerned with the different development programs, particularly in relation to women's development. In addition, our findings agree with similar research on the same issues linked to Bangladesh. Namely, it can be noted that women's decision to work in agriculture is directly related to the distance of the agricultural field from the home, the number of available technologies to use, and the number of available male adults within the family [36].

## 10. Recommendation

Based on the overall study, the following recommendations for policy implications are made:

- Encouraging informal women's groups in rural areas would be a step toward increasing their empowerment, since this would facilitate greater mobility outside the home and their access to media. Local community leaders, extension personnel, NGO workers, and representatives from women's organization would provide a vital contribution to such groups by motivating them toward engaging in various development activities.

- The government can play a significant role in reducing the miserable conditions of women of rural areas by distributing more dairy cattle or leasing land or chicks, free of cost or with lower cost, including limited or no interest.

- More awareness programs and services should be provided by different NGOs and other institutions to encourage rural women to boost their empowerment.

**Author Contributions:** M.S.K. provided data for Tables 1–3, conducted the interviews and all statistical analyses. M.R.M. made the conceptual framework of the study, supervised the research and co-wrote the paper. All authors reviewed the manuscript and revised it several times.

**Acknowledgments:** The research was supported by the Ministry of Education, Science and Technological Development, the Republic of Serbia (Grant III 47009 and 179015) and Bangladesh Agricultural University.

**Conflicts of Interest:** The authors declare no conflict of interest.

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
