# Peer review of "The Determinants of Income of Rural Women in Bangladesh"

_sustainability, doi:10.3390/su11205842_

Round 1

Reviewer 1 Report

. Which is the way in which the Problem Confrontation Index is calculated?

- The sources for the data,  should have to be described more precisely.

- Which  are the main training facilities? 

Author Response

Thank you very much for your comments.We replied on it.

best regards!

Reviewer 2 Report

Dear Authors,

very compliments for your paper intitled "Economy, Ecology and Equity: Closing the Gender Gap in Rural Bangladesh". I found it interesting and innovative, but it need to be improved.

First, basing on its title, you should consider more the ecological aspect of the topic of the paper. For this reason, I suggest you to increase the references considered on ecology and agriculture (I putted only one suggested reference in the text, however you must consider more, in my opinion). Moreover, you must rearrange all the reference numbers in the text as indicate by me with several notes.

Several changes and corrections are indicated in the text.

After the above indicated corrections, the paper could continue in its publication process.

Best Regards.

Author Response

Dear Reviewer,

Thanks a lot for your suggestions.They help us to improve the article very much.Especially thank you for entered comments in the text.

Seven more new references are added and  all technical corrections are done.

I replied  on your questions in track to see what is changed.

Kind regards!

Reviewer 3 Report

The paper deals with a very interesting issue. The literature overview is sufficient. Data and methods of study should be described more detailed, e.g. what was the reason to choose this district for the research? Why there were not chosen more districts to compare them? Maybe it will be interesting for foreigners to have more basic information on the agriculture and agricultural condition in Bangladesh. Why there were only 80 respondents? It is not too sufficient for regression analysis. Who are the respondents? How were the independent variables for model selected?  Why these ones? Explain the reasons to select these variables. How was the income measured? In generally, the level of significance is marked usually like this: “*** Significant at 1%; ** Significant at 5% and * Significant at 10%”. I recommend using the symbols that are usual in the scientific papers. Why p-value 0.000 is marked as *** if you use * for significant at 1%? The interpretations of regression coefficients are wrong. This part should be overwriting. Moreover, it will be better to use econometric models for presentation of relations between income and selected factors. In table 3, there are mentioned only 8 problems? How were the problems selected? Do not consider the respondents any other problems? Is the weight of all problems similar?     

Author Response

Dear Reviewer,

Thanks for your useful comments.We tried to reply on them .I hope ,that our respond will satisfy you.

Best regards!

Authors

Reviewer 4 Report

The title of the paper seems interesting. However, the authors did not fully address the topic well. The paper basically focuses on determinants of rural women income and challenges. The title of the paper should have been "Determinants of incomes of rural women in...".

Both the introduction and theoretical review do not outline the current knowledge, knowledge gap and the contributions of the paper to narrowing that gap. 

The methodology is lacking a conceptual framework. The sampling technique, description of study area as well as model specification needs to be well written. Although the authors mentioned of using both qualitative and quantitative data, there is no evidence of how the data was analysed. 

The results in Table 2 are wrongly interpreted. The authors are interpreting the coefficients as partial elasticities. However, the variables were not transformed into natural logarithms. Moreover, there are inconsistencies with the t-values and statistical significance in Table 2. For example, the t-value of family size (x2) is 9.99 which is wrong. It is supposed to be 0.68. The t-values of Input availability (X5), Decision making ability (X7) and Training and experience (X8) are all wrong. Since the results in Table 2 and their interpretations are faulty, the entire discussion and conclusion are faulty and misleading.

Author Response

Dear Reviewer,

Thanks for your time and useful comments.We tried to reply on all your questions.I hope that you will be more satisfy after our revision.

Please,see the attachment.

Best regards,

Authors

Round 2

Reviewer 1 Report

The problem confrontation index should have to be explained more clearly, and further details in the description should have to be introduced. Why the model is described with a linear function? Data description should have to be improved deeply. The description of the variables should have to be improved, and also the explanation of data sources for the different variables. Why a Cobb-Douglas function has been chosen for the description of a rural model? Descriptive statistics for the different variables should have to be introduced (mean, standard deviation, mode). 

Author Response

Dear Reviewer,

Sorry we submitted our letter but without attachment.

So, we are sending our the revised version of our article.

Please,see below the attachment

Reviewer 2 Report

Dear Authors,

I can see that you improved very much your paper, also using my suggestions and corrections. It is more fine now, in my opinion. However, You could make some little corrections and adjustments forgotten in your revision. You can see what I am writing to you in the revised manuscript.

Many thanks for trust me.

Best wishes.

Author Response

Dear Reviewer,

Technical editing was done according to your latest suggestions.They contributed very much to the quality of the article.

Thank you very much for your time and professional work.

By the way ,one reviewer proposed new title of the paper. Please,see the attachment.

Best regards,

Mirjana

Reviewer 3 Report

I recommend the paper for publication after some formal corrections, e.g. lines 168 - 195 and lines 207-234 are equal.

Author Response

Dear Reviewer,

Thanks a lot for excellent comment.I deleted doubled  paragraph according to your recommendation.One of reviewers suggested to replace the title of the paper with new one  .

Please, find attached the latest version of the paper with new title.

Kind regards,

Mirjana

Reviewer 4 Report

General Comments

Although the authors have tried to improve the paper, the topic is not sufficiently justified by the content. There are still major issues which have not been addressed by the authors. As I indicated in my earlier comments, based on the content and objective of the paper, the title should reads as "The determinants of incomes of rural women in Bangladesh" since the paper is primarily looking at the factors influencing the incomes and challenges faced by rural women. 

Abstract

The abstract needs to be seriously revised based on the provided in the various sections.

Introduction

The introduction still need a thorough revision to meet the journal standard. The authors should clearly indicate the motivation of the paper, knowledge gap and contributions to the paper. These are still not clearly in indicated in the revised version. 

Methods

The methodology section still needs a major revision. The authors claim to use descriptive statistics, functional analysis, the and Cobb-Douglas function as indicated in the abstract and methodology section, there are no evidence of these methods in the paper. Instead, they applied a linear multiple regression. In the results, there are not descriptive results such as the means, standard deviations, etc of the variables used in the regression. The revised version stills lack conceptual framework with no theoretical foundation. Again, the authors are claiming of selecting an appropriate functional form, there is no evidence of functional test in the paper. I kindly suggest that the authors take their time to work on the paper by following these outlines for the methodology.

Conceptual framework

Empirical model specification

Source of data

Results and Discussion

The entire results and discussion needs to be rewritten for the they are not well written. The results are not interpreted well. For example, the authors interpreted the coefficients of the regression as partial elasticity but they are not. They should be rather interpreted as absolute effects. I kindly suggest the following structure for the results and discussion section:

Descriptive results - Here generate summary statistics of the variables both dependent and independent variables included in the model. You can also include other relevant results here.

Determinants of rural women income

Here include results on functional tests, multicollinearity, heteroskedasticity and how you address these econometric issues if they are present. 

Challenges

Include results on the challenges here

Discussion

Take your time to discuss the results here and position the findings within existing literature.

Conclusion

Kindly revise the entire conclusions

Author Response

Hello reviewer,

We did our best to improve our manuscript.I hope that you will more satisfy with this latest version.

Please,see the attachment!

Regards,

Authors

Round 3

Reviewer 1 Report

I think this draft can be accepted.

Author Response

Dear Reviewer,

Thank you for all your suggestions.

Best,

Authors

Reviewer 4 Report

There is little improvement in the revised version but the comments below have not been addressed in the revised version. I encourage the authors to revise the methodology and results/discussion according to these suggestions.

Methods

The methodology section still needs a major revision. The authors claim to use descriptive statistics, functional analysis, the and Cobb-Douglas function as indicated in the abstract and methodology section, there are no evidence of these methods in the paper. Instead, they applied a linear multiple regression. In the results, there are not descriptive results such as the means, standard deviations, etc of the variables used in the regression. The revised version stills lack conceptual framework with no theoretical foundation. Again, the authors are claiming of selecting an appropriate functional form, there is no evidence of functional test in the paper. I kindly suggest that the authors take their time to work on the paper by following these outlines for the methodology.

Conceptual framework

Empirical model specification

Source of data

Results and Discussion

The entire results and discussion needs to be rewritten for the they are not well written. The results are not interpreted well. For example, the authors interpreted the coefficients of the regression as partial elasticity but they are not. They should be rather interpreted as absolute effects. I kindly suggest the following structure for the results and discussion section:

Descriptive results - Here generate summary statistics of the variables both dependent and independent variables included in the model. You can also include other relevant results here.

Determinants of rural women income

Here include results on functional tests, multicollinearity, heteroskedasticity and how you address these econometric issues if they are present. 

Challenges

Include results on the challenges here

Discussion

Take your time to discuss the results here and position the findings within existing literature.

Conclusion

Author Response

Dear Reviewer,

Thanks for your suggestions again.We improved some parts of the article according to your instruction.Firstly ,conceptual framework of the study is added. Also,  added are new subtitles , conclusion is rewritten as recommendation.Several new references are included.Part related to the method is improved.

Our first submission had 13 pages ,but this third one 18 pages.So ,we wrote 5 pages more.

Kind regards!

Authors
